# Structural Relationships between Environmental Factors, Psychological Health, and Academic Performance in Medical Students Engaged in Online Learning during the COVID-19 Pandemic

**DOI:** 10.3390/ijerph20021494

**Published:** 2023-01-13

**Authors:** Ola K. Taleb, Abdullah Sarimah, Ab Hamid Siti-Azrin, Kamarul Aryffin Baharuddin, Ali H. Abusafia

**Affiliations:** 1Biostatistics and Research Methodology Unit, School of Medical Sciences, Universiti Sains Malaysia, Kubang Kerian 16150, Kelantan, Malaysia; 2Department of Emergency Medicine, School of Medical Sciences, Universiti Sains Malaysia, Kubang Kerian 16150, Kelantan, Malaysia; 3Hospital Universiti Sains Malaysia, Kubang Kerian 16150, Kelantan, Malaysia; 4School of Health Sciences, Universiti Sains Malaysia, Kubang Kerian 16150, Kelantan, Malaysia

**Keywords:** academic performance, COVID-19, environmental factor, online learning, psychological health

## Abstract

As a result of the COVID-19 outbreak and the enforced quarantine, universities in Malaysia were required to switch to an online class format. The resulting changes in the environmental factors of students may have had an impact on their psychological health and academic performance. This study aimed to determine the effects of environmental factors and the psychological health of students and examine their structural relationship with academic performance. A cross-sectional design with an online self-reported questionnaire was adopted, and the study was conducted among 207 undergraduate medical students at the Health Campus of Universiti Sains Malaysia. The environmental factors were measured using the lighting–noise–temperature scale and technology scale, while psychological health was assessed using the short version of the General Health Questionnaire and academic performance was determined based on Grade Point Average. Descriptive statistics and structural equation modeling were used for analysis of the data. No significant relationship was found between environmental factors and academic performance, or between environmental factors and psychological health. Nonetheless, the hypothesized structural model provided scientific evidence of an inverse relationship between psychological health and academic performance. These findings could be helpful for academics, health policymakers, and health educators in terms of understanding and promoting psychological wellbeing among university students, as well as improving their academic performance.

## 1. Introduction

The global disruption caused by COVID-19 poses a massive challenge to global health and continues to affect people as well as students worldwide. Borders and all academic institutions were closed, and economic activities came to a standstill due to the implementation of the Movement Control Order (MCO) in Malaysia on 18 March 2020, to keep the spread of COVID-19 and associated mortality under control [1]. The campus closure and movement restriction order had an impact on formal learning. In April 2020, the Malaysian government ordered students to return to their home countries and continue their studies through online learning [2].

Medical students were one of the most severely affected groups, as their lessons at hospital and other health care centers where they learn essential clinical skills had been suspended until further notice. They were also required to adapt quickly to the online learning format, which required much adjustment and was unlike their face-to-face and patient-based learning format in the health care setting. These unfavorable circumstances probably led to an increase in stress levels among medical students, especially those who experienced coping and adjustment difficulties. All these factors may have contributed to depression, anxiety, and stress among medical students.

Mahdy (2020) demonstrated that the lockdown imposed as a result of the COVID-19 pandemic had varying degrees of impact on most students’ academic performance [3], as they were required to adapt themselves to new environmental conditions, such as lighting, noise, temperature (LNT) and technologies, that are different from those at classrooms in their university. These new environmental conditions could be uncomfortable and have an impact on their academic performance [4]. Adequate lighting is important for the learning process. During the day, natural light through windows is available, and it may be augmented with artificial light sources, with the artificial illumination continuing throughout the night [5]. In addition, excessive noise is also detrimental to the teaching and learning process because it is distracting and limits attention and cognition. Specifically, excessive noise can make it difficult for students to hear and understand the teacher’s speech [5,6]. In addition, thermal discomfort caused by elevated classroom temperatures has been demonstrated to affect students’ ability to accomplish standard academic tasks [7,8]. Further, heat stress caused by high outside temperatures has been found to be associated with an increase in the number of students failing tests [9]. Inadequate online learning infrastructure and limited accessibility to the internet also make the online learning process harder for students [10]. Students need to have access to reliable equipment [11], and students with limited access are at a significant disadvantage when compared to students with unlimited access [12].

During the COVID-19 pandemic, psychological health problems, especially depression, anxiety, and stress, have become common among students, especially medical students [13]. In students at higher-education institutions, psychological factors have been found to affect academic achievements as a result of test anxiety, poor performance, low self-confidence, unrealistic worry, and fear or uneasiness that interfere with their ability to function normally [14,15,16]. Several recent studies have suggested that environmental factors may also contribute to the onset or aggravation of psychological health [17]. Lighting is one of the most important factors for a positive learning environment [18], and it has an important role in improving the psychological health of students [19]. There is evidence for the significance of lighting for psychological wellbeing, as well as personal health and wellbeing [20]. For example, Mirrahimi et al. (2013) demonstrated the effects of lighting on the health, psychology, and cognitive abilities of students [19]. In addition, noise has been found to have negative effects on psychological health [21] by causing emotional distress and discomfort [22]. For example, Beutel et al. (2016) demonstrated that depression and generalized anxiety disorder increased dose-dependently with the degree of total noise annoyance [23]. Further, Bourion-Bédès et al. (2021) demonstrated that one of the main risk factors for anxiety among students during lockdown was noise, both inside and outside the house [24], and a significant association between noise exposure and stress levels has also been reported (*p* = 0.023) [25]. Rising temperatures can directly and indirectly cause human pathologies that affect both physical and mental health [26]. The tropical climate in Malaysia is hot and humid, and this environment may have an impact on air temperature, causing dissatisfaction, unhappiness, and discomfort among students and affecting their productivity [27]. 

The switch to the online learning mode has led to severe anxiety among medical students, who have also reported inadequate internet access, which is required for online learning [28]. Happiness does have a statistically significant wealth gradient, with students with internet access being more likely to be happy [29]. According to Haris & Al-Maadeed (2021), the pandemic and resulting transition to the online learning environment has led to mental stress among students due to the absence of a campus environment and connectivity, as well as technical problems [30].

However, no studies in Malaysia so far have focused on the relationship between psychological health and environmental factors (LNT and technology) and how they affect academic performance. Therefore, this research aimed to study the relationship between environmental factors (LNT and technology), psychological health, and academic performance among medical students at the Health Campus of USM during the COVID-19 period in Malaysia. It is important to examine the factors that influence students’ academic performance during the pandemic period, as this can provide insight into how the Malaysian university and government can improve the online learning environment in Malaysia. The hypothesized relationship between environmental factors, psychological health, and academic performance is depicted in Figure 1.

## 2. Materials and Methods

### 2.1. Study Design and Ethical Approval

A cross-sectional study was conducted on 207 undergraduate medical students from the School of Medical Sciences, USM. Second- and third-year undergraduate medical students in academic year 2021–2022 who participated in online learning during the second semester of academic year 2020–2021 were included. Re-sit students and students with a history of underlying psychological issues were excluded. 

The study was approved by the Human Ethics Committee of USM (approval number USM/JEPeM/21080583) and the academic office of the university. This ethical approval allowed the researcher to conduct the research on the target population. Implied consent was obtained from each of the participants after they had received the link to the Google form of the questionnaire. The information provided about the research included the purpose of the study, study procedures, potential risks of participation, and benefits of participation in the study. All participants voluntarily participated in this research study. The researcher guaranteed that the data collected were kept confidential and only used for research purposes. After the survey was completed, the participants were thanked for their involvement in the research study.

Table 1 lists the participants’ demographic information. As shown in the table, 52% of the participants were in their second year, and 73% were women. Most participants (96.1%) were of Malay descent, and these participants continued to live on campus while learning online. Laptops were the primary digital device for 62.8%, who learned individually. Further, 88.4% of the participants used Wi-Fi to attend the online courses.

With regard to the classification of household income, there are three classes of household income in Malaysia: B40, M40, and T20. B40 represents the bottom 40% of the average Malaysian household income and corresponds to less than RM4850 per month. M40 represents the middle 40% and corresponds to RM4851–RM10970 per month. T20 represents the top 20% of Malaysian household income and corresponds to RM10971 per month. In this study, 42% of participants were classified under M40, that is, they had a family income of RM4851 to RM10970. 

### 2.2. Sample Size 

The sample size was calculated based on the method of Kline (2011) [31]. He suggested, using structural equation modeling (SEM), that the acceptable sample size for studies is about 200 cases. Considering a 20% dropout rate, the minimum required sample size for the study was 250 students. A convenience sampling method was applied to select participants based on the inclusion and exclusion criteria. 

### 2.3. Data Collection

Data were collected using an online self-administered questionnaire. The link to the Google Form of the questionnaire was sent to the class leader of each year through WhatsApp and email. They shared the online survey link with other students. Participants completed the online survey in 20–30 min. They were informed that it was compulsory to answer all the questions. The GPA reported by the students in the survey was verified with the results provided by the academic office.

This questionnaire consists of three parts: (i) 9-item proforma form (about sociodemographic profiles and GPA for the second semester of the academic year 2020/2021 examination), (ii) 15-item questionnaire on environmental factors (LNT and technology questionnaires) and (iii) 12-item psychological health questionnaire (GHQ-12).

### 2.4. Instruments

#### 2.4.1. The Environmental Questionnaire

##### LNT Scale

The LNT scale was developed by Realyvásquez-Vargas et al. (2020) [4] and consists of nine items that are used to assess the environmental factors that impact students’ online learning in three domains, namely, lighting, noise, and temperature. The English version of LNT was found to have a good fit based on the following values: comparative fit index (CFI) = 0.99, Tucker–Lewis index (TLI) = 0.98, standardized root mean square residual (SRMR) = 0.03, and root mean square error of approximation (RMSEA) = 0.03 (90% confidence interval [CI] = 0.00–0.07). For lighting, noise, and temperature, the composite reliability (CR) values were 0.81, 0.81, and 0.84, respectively, and the average variance extracted (AVE) values were 0.61, 0.59, and 0.63, respectively [11].

The present study shows that the measurement model had a good fit based on recommended fit indices without modification indices. The results of robust fit indices were as follows: CFI = 0.99, TLI = 0.98, SRMR = 0.03, and RMSEA = 0.03. The factor loadings ranged from 0.74 to 0.82 (*p*-value < 0.001). The CR for lighting, noise, and temperature was 0.82, 0.81, and 0.84, respectively. The AVE for lighting, noise, and temperature was 0.61, 0.59, and 0.63, respectively.

The tool uses a five-point Likert scale, where 1 = never, 2 = hardly ever, 3 = sometimes, 4 = usually, and 5 = always. The first domain is lighting (three items), which measures the effects of indoor lighting on students’ academic performance during online classes. The second domain is noise (three items), which measures the effects of noise pollution on students’ academic performance during online classes. The third domain is temperature (three items), which measures the effects of thermal comfort on students’ academic performance during online classes.

##### Technology Scale

The technology scale was developed by Abou Naaj et al. (2012) [11] and consists of six items that measure the adequacy of technology that impacts students’ academic performance during online classes. It uses a five-point Likert scale, where 1 = strongly disagree, 2 = disagree, 3 = neither agree nor disagree, 4 = agree, and 5 = strongly agree. The English version of the technology scale showed a good fit to the data: CFI = 0.99, TLI = 0.97, SRMR = 0.02, RMSEA = 0.06 (95% CI = 0.00–0.16). The CR was 0.84 and AVE was 0.51 [11].

The present study showed that the model had a good fit based on the values of all indices, except for the upper 90% CI of robust RMSEA, which was 0.16. These were the values for the remaining robust fit indices: CFI = 0.99, TLI = 0.97, SRMR = 0.02, and RMSEA = 0.06. The factor loadings ranged from 0.62 to 0.79 (*p*-value < 0.001). The CR for technology was 0.84, and the AVE for technology was 0.51. Figure 2 shows error covariances among the indicators for technology.

##### GHQ-12 Scale

The 12-item GHQ-12 scale provides a quantitative measure of psychological health. The GHQ is a screening instrument that was developed to determine the severity of psychological distress that an individual has experienced during the previous 6 months using a four-point scale ranging from 0 (more than usual) to 3 (much less than usual). It consists of 12 items that describe mood states, with six of them positively phrased (1, 2, 4, 7, 8, and 12) and six of them negatively phrased (2, 5, 6, 9, 10, and 11) [32]. According to the standard GHQ scoring method (GHQ–0011), GHQ items are scored dichotomously. Score 0 and 1 was replaced to Score 0, whereas Score 2 and 3 was replaced to 1. The scores range from 0 to 12, with a total score of 4 and above indicating psychological problems. The English version of GHQ-12 has good validity [33].

### 2.5. Statistical Analysis

IBM SPSS Statistics version 27.0 (IBM Corp., Armonk, NY, USA) and the Lavaan package of R Studio were used for data analysis. Descriptive statistics of the numerical variables are presented as mean and standard deviation (SD), while categorical variables are presented as frequency and percentage. Further, inferential statistics were analyzed with SEM, which was used to examine the structural relationship between environmental factors, psychological health, and academic performance. Due to the violation of normality, the maximum likelihood estimator (MLR) was used in the analysis. The items in the measurement model for psychological health were parceled into one domain (psychological health), while the measurement models LNT and Technology were maintained.

Fit indices were used to evaluate the model’s fitness. RMSEA < 0.08, SRMR < 0.08, TLI > 0.95, and CFI > 0.95 [31,34]. Significant paths (β) with the corresponding 95% CI, critical ratio, and statistical significance values (<0.05) are presented in this study. 

## 3. Results

### 3.1. Descriptive Statistics of Constructs

The items in the measurement model for psychological health were parceled into one domain, named psychological health, and were treated as observed items. The descriptive statistics of the constructs are shown in Table 2.

### 3.2. LNT Scale

Nine items in Table 3 were applied to assess environmental factors (LNT) related to the online classes that affect academic performance. More than 80% of the students reported that they were satisfied with the lighting, and 70% were satisfied with the temperature. However, only half of the students stated that they were able to tolerate noise. 

### 3.3. Technology Scale

The six items presented in Table 4 were applied to assess the technical quality of online classes that affect their academic performance. Only 57.8% of students were satisfied, while 15% encountered technical problems.

### 3.4. Psychological Health Scale

The twelve items presented in Table 5 were applied to assess the psychological health of participants and how it affected their academic performance. According to the results, 52.7% of students had a total score of less than 4, while 47.3% had a total score more than 4.

### 3.5. Initial Structural Model

The environmental factors (LNT and technology), psychological health, and academic performance were imputed into the proposed structural model using the estimator MLR, in order to examine their relationship. Table 6 shows the results from the initial model (model 1) which showed that the data had a good fit (CFI = 0.96, TLI = 0.95, SRMR = 0.05, RMSEA (90% CI) = 0.04 (0.02–0.06). The value of chi-square (degree of freedom) was 137 (102). The results met the recommended value. Therefore, no modification was required, and model 1 was considered as the final model (Figure 2). Model 1 was tested, and the results showed that only one out of nine of the hypothesized relationships was significant. The path relationships in the final model are summarized in Table 7, which differentiates between significant and non-significant paths based on the SEM results. The results from Table 7 show that there was an inverse relationship between psychological health scores and academic performance based on GPA (*β* = −0.21, *p* = 0.004).

## 4. Discussion 

### 4.1. Online Learning

A virtual learning environment (VLE) is a set of teaching and learning tools designed to enhance a student’s learning experience by including computers and the internet in the learning process. The principal components of a VLE package include curriculum mapping (breaking curriculum into sections that can be assigned and assessed), student tracking, online support for both teacher and student, electronic communication (e-mail, threaded discussions, chat, and web publishing), and internet links to outside curriculum resources. In general, VLE users are assigned either a teacher ID or a student ID. The teacher sees what a student sees, but the teacher has additional user rights to create or modify curriculum content and track student performance. There are a number of commercial VLE software packages available, including Blackboard, WebCT, Lotus LearningSpace, and COSE.

In view of the COVID-19 pandemic and the MCO imposed by the government, a blended learning substitution model was adopted. With this model, students learn through online classes as well as traditional face-to-face lessons. Online class learning is composed of asynchronous and synchronous methods. 

Asynchronous methods include online teaching and learning strategies that do not involve real-time interaction, for example, recorded videos/lectures and self-guided learning materials. The Elearn@USM platform is the university’s official e-learning platform used for asynchronous online teaching and learning. Both lecturer and student can access this learning platform. 

Synchronous online teaching and learning methods involve online teaching and learning in real time. Lecturers use online meeting platforms, such as Webex, Microsoft Team, Zoom, Google Meet, and others, to conduct teaching sessions with students. The students can interact or ask questions, and lecturers can respond in real time.

The COVID-19 pandemic and lockdown had varying degrees of effects on most students [3]. Some students experienced isolation and loss of interest after participating in online learning for a prolonged period. They also experienced fatigue as a result of sleeping problems, technological problems, and rapid system and environmental changes. Several recent studies have suggested that environmental factors may contribute to the onset or aggravation of psychological health [27]. 

### 4.2. Environmental Factors (LNT and Technology), Psychological Health, and Academic Performance

In the current study, the overall mean (SD) scores for the lighting and noise scales were 12.78 (2.13) and 10.19 (2.76), respectively. A study by Realyvásquez-Vargas et al. (2020) on Mexican students which used the same scales reported that the overall mean score of lighting and noise were 11.88 and 9.95, respectively [4]. These results indicate that the USM students in this cohort had better lighting and noise conditions than the Mexican students from the previous study. This is probably because most students from USM lived within the campus, and the university provides good conditions for student learning. In contrast, the overall mean (SD) for the temperature scale was 11.79 (2.41) in this study and 11.48 in the Realyvásquez-Vargas et al. (2020) study [4]. This implies that the Mexican students had better temperature conditions than the USM students. This is also probably because most students from USM lived within the campus in rooms that did not have air conditioning. In addition, the tropical hot and humid climate of Malaysia may also have contributed to this score. This environment may have caused discomfort and, therefore, dissatisfaction unhappiness among students, and affected their productivity [27].

The overall mean score (SD) for the technology scale was 23.88 (3.47). Compared to study by Selvanathan et al. (2020), which was also conducted among students in Malaysia’s universities and used the same scale, the overall mean score for the technology scale was 3.66 and the overall mean score was 20.22 [35]. Although both studies were conducted in Malaysia, the current study has a higher mean score than the Selvanathan et al. (2020) study. This difference is probably because data collection in the previous study was done when the students were living at home. Therefore, the scores may be related to the socio-economic status of their families and the quality of the technical issues. In contrast, in the current study, most of the students stayed within the campus, where they have similar access to technology.

In the current study, the mean score (SD) for psychological health was 4.16 (3.92). A score of more than 4 indicates that the participant has psychological problems. Therefore, the mean score implies that most undergraduate medical students at USM have psychological problems. This result was somewhat similar to that for medical students from Suez Canal University, Egypt, who had a mean (SD) score of 4.7 (3.1). However, the score in the present cohort was higher than that reported in previous studies on undergraduate students (3.12 (3.32)) from the faculties of medicine, dentistry, and health sciences on the Health Campus of USM [36].

The interrelationship between environmental factors, psychological health, and academic performance were examined with SEM. The structural framework was built on a conceptual framework based on previous theoretical and empirical studies. The findings of the structural model analysis revealed that only one of the nine hypothesized relationships was significant. That is, in this study, psychological health scores were found to have an inverse relationship with academic performance (indicated by GPA). The findings showed that students with higher scores for psychological health were likely to have lower GPAs. This supports the findings of previous studies which showed that higher scores for psychological health were associated with lower GPAs [14,15,37,38]. Similarly, in a longitudinal study conducted among Emirati students to determine the relationship between mental health and students’ academic performance, higher Patient Health Questionnaire (PHQ-9) scores were associated with lower current GPAs (r_s_  =  −0.171, *n*  =  397, *p*  <  0.001) [37]. In addition, a cross-sectional study conducted among Malaysian medical colleges reported that stress, anxiety, and depression have a negative impact on the learning and academic performance of university students [38]. 

Further, Bas (2021) demonstrated a positive relationship between mental health and academic performance [39], and Chattu et al. (2002) also showed a significant association between psychological health and the academic performance of students [14]. According to Duffy et al. (2020), depressive symptoms cause a decrease in GPA [16]. Further, Beharu (2018) showed that psychological factors, such as stress, anxiety, depression, lack of motivation, loneliness, helplessness, and phobias, in students in higher-education institutions can affect their academic achievements by causing test anxiety, poor performance, low self-confidence, unrealistic worry, and fear or uneasiness, which interfere with their ability to function normally [15]. Along the same line, in Bas’s (2021) meta-analysis, a positive relationship was found between mental health and academic performance [39]. According to Mahdy (2020), the lockdown had variable degrees of impact on most students’ academic performance [3]. For example, loneliness, loss of interest, and fatigue are commonly reported after online learning for a prolonged period. Psychological health may influence academic performance in a variety of ways, as those with psychological problems may find interactions difficult and may fail to engage in their classes [40].

In the present study, environmental factors (LNT and technology) did not have a significant relationship with academic performance. This is because most of medical students had good score in academic performance. The students learned through Elearn@USM platform as an asynchronous online learning. Asynchronous learning includes recorded videos/lectures, self-guided learning materials, etc. If the students had any problems regarding their subjects, they can look back at the recorded videos, uploaded by their lecturers. 

However, this contradicts the findings of previous studies [3,41,42,43,44,45,46]. For example, Mogas-Recalde & Palau (2021) confirmed the direct impact of classroom lighting on student performance (concentration, attentiveness, achievement, etc.) [41], and Brink et al. (2021) showed that the indoor environmental quality, which includes indoor air, as well as thermal, acoustic, and lighting conditions, can contribute positively to the quality of learning and short-term academic performance of students [42]. 

In Mexico, a study was conducted by Realyvásquez-Vargas et al. (2020) to explore whether environmental factors (LNT) affected students’ academic performance during COVID-19 [4]. They found that students’ academic performance was indeed affected by environmental factors during the COVID-19 pandemic. That is, students with better environmental conditions were found to have better academic performance according to the QEOC questionnaire. Similarly, Norazman et al. (2018) discovered that the quality of lighting in the classroom, whether natural or artificial, has an impact on students’ achievement, motivation, attendance, skills, concentration, and focus throughout the day [43], and Samani & Samani (2012) showed that there was a significant relationship between lighting quality and student’s learning achievement [44]. Along the same line, Gonzales et al. (2018) illustrated how intermittent internet disconnection problems negatively affected the GPA of students [45]. Thus, adequate internet services enhance access to information required by the students, and this has a positive effect on their academic achievement [46].

The present findings indicated that environmental factors (LNT and technology) do not have a significant relationship with psychological health, but these findings are also contrary to those of previous studies [21,24,25,26,28,29,30]. This difference in results is probably attributable to the living conditions of the present cohort, as most of them lived within the campus and were exposed to the same environmental factors. Thus, the incorrect conclusion that the independent variable has no significance could have been the result of a ceiling effect [46]. The ceiling effect occurs when a variable’s scores approach their maximum [47]. Pilot testing is the best solution to resolve this issue, since it allows for the problem to be discovered early [48]. Another method is to collapse categories that distribute data more evenly in order to decrease the floor or ceiling impact. This strategy, however, may result in loss of information and power, and this could change the implication of many health-related outcomes with a well-defined range of possible responses [49]. In addition, adjusted weight least square mean and variances or MLR with numerical integration can be applied to analyze ordered categorical data with the ceiling effect [50]. The researchers in the current chose MLR as the estimator. As only one of the hypothesized relationships was significant, the indirect effect and interaction were not analyzed. 

To the best of the researcher’s knowledge, this is the first study to use SEM to examine the relationship between environmental factors (LNT and technology), psychological health, and academic performance among Malaysian students. SEM was used to investigate the multiple interrelationships between variables in the hypothesized structural model. As a result, the type 1 error for the study is assumed to be below. In addition, the questionnaires used were valid and reliable, and the GPA was verified by the academic office. However, there are a few limitations that need to be addressed. The data were collected only from a single college and university, and this may have led to a ceiling effect as most students were exposed to the same environmental factors. In addition, it limits the generalizability of the results to other university students. Due to time constraints, data re-collection to anticipate the problem arising from the ceiling effect was not conducted. The convenience sampling method was applied, instead of random sampling, because of the online format of the questionnaires, the large sample size needed for this study, and the time constraint. As online self-administered questionnaires were used for data collection, there is a possibility of bias arising from insincere or dishonest responses. Furthermore, the study had a cross-sectional design. As a result, the exposure and the outcome were evaluated concurrently, and this made it impossible to distinguish between the cause-and-effect factors. Thus, it was not possible to determine the direction of effect due to a lack of evidence to establish a temporal association between the exposure and the outcome. For example, psychological health scores were found to be significantly and negatively associated with GPA. However, the data do not shed light on whether psychological health problems lead to low GPA, or vice versa. Moreover, the researcher did not include internet speed issue as a technology-related item. It is recommended that this factor be included as a technology-related item in future studies on this topic.

## 5. Conclusions

The present study demonstrated an inverse relationship between psychological health scores and academic performance. In contrast, environmental factors were not found to be associated with psychological health or academic performance. Therefore, the environmental factors light, noise, temperature, and technology were not found to have a direct or indirect effect on academic performance.

## Figures and Tables

**Figure 1 ijerph-20-01494-f001:**
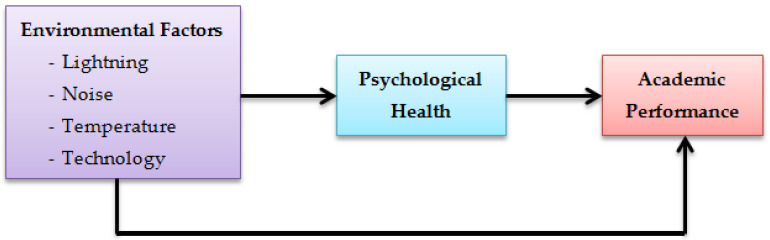
Hypothesized model of constructs.

**Figure 2 ijerph-20-01494-f002:**
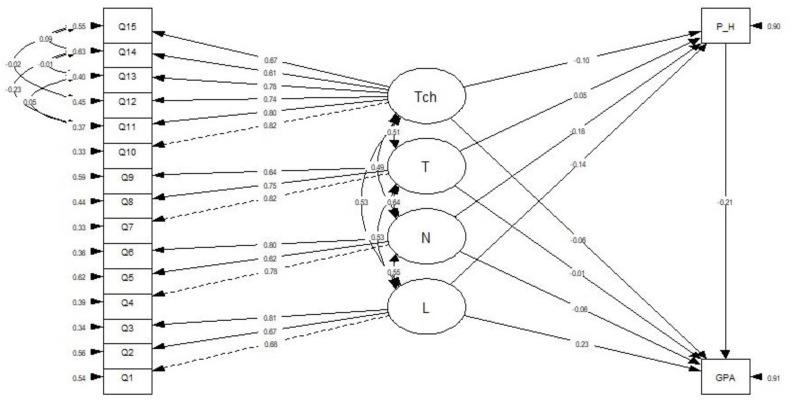
Final SEM Model (model 1). Tch = Technology, T = Temperature, N = Noise, L = Lighting, P_H = Psychological health, GPA = Grade Point Average (academic performance).

**Table 1 ijerph-20-01494-t001:** Demographic characteristics of the participants (n = 207).

Characteristics		Frequency (%)
Gender	Woman Man	152 (73.4) 55 (26.6)
Ethnicity	Malay Chinese Indian Arab Others	162 (78.3) 22 (10.6) 17 (8.2) 3 (1.4) 3 (1.4)
Academic years	2nd year 3rd year	108 (52.2) 99 (42.8)
Family income (RM)	<4850 4851–10,970 >10,971	73 (35.3) 87 (42.0) 47 (22.7)
Mode to access online classes (most of use)	Wi-Fi Mobile data Internet cafe	183 (88.4) 24 (11.6) 0 (0.0)
Current accommodation	Inside campus Urban Rural	199 (96.1) 4 (1.9) 4 (1.9)
Digital tools	Laptop Mobile phone Ipad/Tablet Desktop	130 (62.8) 21 (10.1) 53 (25.6) 3 (1.4)

**Table 2 ijerph-20-01494-t002:** The descriptive statistics of constructs (n = 207).

Construct	Range	Overall Mean (SD) Items	Overall Mean (SD) Scale
Lighting	3–15	4.26 (0.71)	12.78 (2.13)
Noise	3–15	3.52 (0.91)	10.19 (2.73)
Temperature	3–15	3.93 (0.80)	11.79 (2.41)
Technology	6–30	3.98 (0.58)	23.88 (3.47)
Psychological health	0–12	-	4.16 (3.92)
GPA	0–4	-	3.51 (0.32)

**Table 3 ijerph-20-01494-t003:** Summary of item characteristics for LNT (n = 207).

Items	Mean (SD)	Median (IQR)	Score, n (%)				
			1	2	3	4	5
Q1: The level of lighting in my study area allows me to see clearly what is around.	4.28 (0.87)	4.0 (1.0)	1 (0.5)	2 (1.0)	31 (15.0)	78 (37.7)	95 (45.9)
Q2: I can control the level of lighting in my study area when taking online classes	4.26 (0.98)	5.0 (1.0)	4 (1.9)	10 4.8)	24 (11.6)	59 (28.5)	110 (53.1)
Q3: The level of lighting (from lamps, computer screen) in my study area allows me to have visual comfort	4.25 (0.83)	4.0 (1.0)	1 (0.5)	4 (1.9)	34 (16.4)	72 (34.8)	96 (46.4)
Q4: I have privacy in my study area when taking classes online	3.41 (1.15)	4.0 (2.0)	10 (4.8)	27 (13.0)	62 (30.0)	52 (25.1)	56 (27.1)
Q5: The noise level (coming from devices, people’s talks, external sources) in my study area allows me to concentrate	3.46 (1.01)	3.0 (1.0)	7 (3.4)	22 (10.6)	82 (39.6)	60 (29.0)	36 (17.4)
Q6: I can control the noise level in my study area	3.53 (1.12)	4.0 (1.0)	6 (2.9)	35 (16.9)	60 (29.0)	55 (26.6)	51 (24.6)
Q7: The temperature in my study area allows me to be comfortable and concentrate	3.93 (0.95)	4.0 (2.0)	3 (1.4)	13 (6.3)	44 (21.3)	83 (40.1)	64 (30.9)
Q8: I can control the temperature in my study area	3.79 (1.08)	4.0 (2.0)	5 (2.4)	25 (12.1)	42 (20.3)	71 (34.3)	64 (30.9)
Q9: The air quality in my study area is appropriate	4.07 (0.87)	4.0 (1.0)	2 (1.0)	6 (2.9)	41 (19.8)	84 (40.6)	74 (35.7)

1 = never, 2 = hardly ever, 3 = sometimes, 4 = usually, 5 = always.

**Table 4 ijerph-20-01494-t004:** Summary of item characteristics for technology (n = 207).

Items	Mean (SD)	Median (IQR)	Score, n (%)				
			1	2	3	4	5
Q10: The instructor’s voice is audible	4.02 (0.70)	4.0 (1.0)	0.0 (0.0)	4 (1.9)	36 (17.4)	119 (57.5)	48 23.2)
Q11: Course content shown or displayed on the smart board is clear	4.17 (0.65)	4.0 (1.0)	0.0 (0.0)	2 (1.0)	23 (11.1)	119 (57.5)	63 (30.4)
Q12: The microphone is in good working condition	4.06 (0.76)	4.0 (1.0)	0.0 (0.0)	6 (2.9)	3.6 (17.4)	105 (50.7)	60 (29.0)
Q13: The video image is clear and comprehensive	4.02 (0.71)	4.0 (1.0)	0.0 (0.0)	5 (2.4)	35 (16.9)	117 (56.5)	50 (24.2)
Q14: Technical problems are not frequent, and they do not adversely affect my understanding of the course	3.57 (0.96)	4 (1.0)	5 (2.4)	26 (12.6)	51 (24.6)	96 (46.4)	29 (14.0)
Q15: The technology used for online teaching is reliable	4.04 (0.68)	4.0 (1.0)	0.0 (0.0)	4 (1.9)	32 (15.5)	123 (59.4)	48 (23.2)

1 = strongly disagree, 2 = disagree, 3 = neither agree nor disagree, 4 = agree, 5 = strongly agree.

**Table 5 ijerph-20-01494-t005:** Summary of items descriptive of psychological health (n = 207).

Items	0 n (%)	1 n (%)
Q16: Been able to concentrate on what you’re doing?	125 (60.4)	82 (39.6)
Q17: Lost much sleep over worry?	126 (60.9)	81 (39.1)
Q18: Felt you were playing a useful part in things?	145 (70.0)	62 (30.0)
Q19: Felt capable of making decisions about things?	163 (78.7)	44 (21.3)
Q20: Felt constantly under strain?	110 (53.1)	97(46.9)
Q21: Felt you couldn’t overcome your difficulties?	115 (55.6)	92 (44.4)
Q22: Been able to enjoy your normal day-to-day activities?	138 (66.7)	69 (33.3)
Q23: Been able to face up to your problems?	151 (72.9)	56 (27.1)
Q24: Been feeling unhappy and depressed?	130 (62.8)	77 (37.2)
Q25: Been losing confidence in yourself?	117 (56.5)	90 (43.5)
Q26: Been thinking of yourself as a worthless person?	141 (68.1)	66 (31.9)
Q27: Been feeling reasonably happy, all things considered	161 (77.8)	46 (22.2)

0 = Don’t have a problem, 1 = Have a problem.

**Table 6 ijerph-20-01494-t006:** Model fit indices of the initial structural model.

Model	CFI	TLI	SRMR	RMSEA (90% CI)
Model 1	0.96	0.95	0.05	0.04 (0.02, 0.06)

CFI = comparative fit index, TLI = Tucker–Lewis index, SRMR = Standardized Root Mean square Residual, RMSEA = Root Mean Square Error of Approximation, CI = Confidence Interval.

**Table 7 ijerph-20-01494-t007:** Hypothesized path relationships in the final model (model-1).

Hypothesis	Pathways	Β (95% CI)	CR	*p*-Value
H_1_	GPA ß L	0.23 (−0.02, 0.49)	1.63	0.076
H_2_	GPA ß N	−0.06 (−0.29, 0.17)	0.52	0.599
H_3_	GPA ß T	−0.01 (−0.26, 0.25)	−0.05	0.960
H_4_	GPA ß Tch	−0.06 (−0.23, 0.12)	−0.66	0.513
H_5_	GPA ß P_H	−0.21 (−0.35, −0.06)	−2.83	0.004
H_6_	P_H ß L	−0.14 (−0.35, 0.06)	−1.34	0.177
H_7_	P_H ß N	−0.18 (−0.41, 0.05)	−1.47	0.137
H_8_	P_H ß T	0.04 (−0.19, 0.28)	0.39	0.695
H_9_	P_H ß Tch	−0.09 (−0.28, 0.08)	−1.02	0.307

β = regression weights of pathways, CR = Critical Ratio, L = Lighting, N = Noise, T = Temperature, Tch = Technology, P_H = Psychological health.

## Data Availability

Not applicable.

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
