# Peer review of "Structural Relationships between Environmental Factors, Psychological Health, and Academic Performance in Medical Students Engaged in Online Learning during the COVID-19 Pandemic"

_ijerph, 2023, doi:10.3390/ijerph20021494_

Round 1

Reviewer 1 Report

For the abstract, there is one comment as below:

1. It is rare for authors to state the results in details in the Abstract section. (From line 24 to line 29). So the results are recommended to be deleted.

 For the Introduction, there are two comments as below:

1. Regarding the hypothesized mediating model, psychological health was suggested as the mediator in the link from environmental factors to the academic performance of students (see line 78). However, literature review on the relationship between environmental factors and psychological health is lacking. It must be added. Otherwise, mediation model cannot be established.

2. From line 70, there are two words “studies study”. Please delete one if they are repeated.

 Regarding Material and Methods, there are several comments as below:

1. For the subsection 2.1, please clarify what inclusion requirements were. (see line 86)

2. For the subsection 2.1, there is no information about ethical approval such as student consent. Please add.

3. Descriptive statistics of the participants were lacking in subsection 2.1. Please add.

4. The authors mentioned that the minimum required sample size for the study was 250 students. But the sample in this study was 207 medical students. Does it mean that the study does not have enough sample size for conducting SEM analyses?

5. For subsection 2,2, please clarify what exclusion requirements were. (see line 93)

6. The subsection “Instruments” is suggested to be added before 2.3.1 (see line 105)

7. For the subsection 2.3.1.1, is the information about model fit indices, composite reliability and average variance extracted belonging to the present study? (see lines 109-114). If not, please add the model fit indices, composite reliability and average variance extracted of this study.

8. For the subsection 2.3.1.2, is the information about model fit indices, composite reliability and average variance extracted belonging to the present study? (see lines 126-129). If not, please add the model fit indices, composite reliability and average variance extracted of this study.

For the Results part, there are several comments as below:

9. It is usual to put participants’ demography in the “Materials and Method” section. You may put the information from lines 156 to 160 to the “Materials and Method” section. Besides, you may put Table 1 in the “Materials and Method ” section.

10. Please clarify the difference between overall mean (SD) items and overall mean (SD) scale (see Table 2 on line 166)

11. The authors stated that the results from the initial model (model-1) showed a good fit to the data, CFI = 0.96, TLI = 0.95, SRMR = 0.05, RMSEA 192 (90% CI) = 0.04 (0.02, 0.06). Please indicate the value of chi-square and degree of freedom. Besides, please indicate the fit indices for the measurement model containing all environment factors, psychological health and academic performance.

12. For Figure 2 in line 207, there are error covariances among the indicators for Tch. Briefly describe the presence of error covariances in the “Instruments” section.

 For the Discussion part, there are two comments as below:

13. Even though only one inverse relationship between psychological health and academic performance of students was found in this study, the results of mediating effects of psychological health in the links from environmental factors to academic performance must be stated in the paper. It is because the conceptual model to be tested is the mediation model.

14. Although the results of mediating effect were insignificant, relevant literature must be added to in the Discussion section to illustrate and discuss the possible inter-relationship among three variables (environmental factors, psychological health and academic performance).

Reviewer 2 Report

In this study, the authors investigate the relationship between environmental conditions, psychological well-being and academic results of university students. They find no significant relationship between environmental conditions, well-being and academic performance, but highlight an inverse relatioship between psychological well-being and academic performance.

This article is well-structured, with an overall conciseness and clarity of scopes, objectives, and result presentation and discussion. The results only marginally add to an already well-established field, but they are well contextualized and discussed, making for an enjoyable read and an overall worthy contribution to the literature.

Materials and methods are well-described; the results make good use of tabs and graphics; the bibliography is up to date and relevant, and the language is very clear.

I have only one minor methodological issue. In the paper, it is said that the final sample consisted of 207 students, of whom 199 (96.1%) lived in the campus. Since this condition is so prevalent, I would advise to exclude from the analysis the few students not living in the campus in order to have a more homogeneous sample. Since the main finding of the study (no relationship between environmental factors and students perceived health and performance) is related by the authors to the majority of the students living in campus (contrary to other studies cited), this would in my opinion improve the quality of sampling and analysis.

There is also a little formatting error in table 3 (the first line is bold).

Reviewer 3 Report

Dear Authors,

Thank you for submitting your manuscripts. 

Unfortunately, I found the design, English writing, presentation and interpretation of the results rather poor.  Literature Review rambles from topic to topic without a clear focus. Discussion mixes results with trivial studies without making distinctions about quality or relevance.

Please find some more specific comments below:

- Abstract needs to be reconstructed as it repeats information in its results section,

- What was the VLE used in this study?

- Why were psychological health problems more common amongst medical students during COVID? 

- In conclusion - it was understood that there was a relationship between between environmental factors with psychological health and academic performance but it was not significant, but it was stated otherwise. 

- Ethical approval was not discussed and was only included within a sub-title. Hence the questionnaire analysed sensitive info such as psychological disorders, it is important to address how, where and by whom data was gathered and stored and for how long. 

- What does family income of between 158 RM4851 and RM10970 mean?

- Line 157: "Most participants (96.1%) were Malay and stayed in the campus." Where exactly and how they studied online on campus? As groups or individually? 

- Line 163-164: the sentence is not clear and difficult to understand!

- Line 174-177: Results from section 3.2  is repeated in section 3.3. 

- Were there any VLE access or the internet speed issue as technology related items? These are not studied nor discussed. 

Best Wishes!

Round 2

Reviewer 1 Report

Thank you for your effort for making amendment on the manuscript. Here are some minor questions that I need your clarification:

1. From lines 104-106, you state that "However, no studies so far have focused on the relationship between psychological health and environmental factors (LNT and technology) and how they affect academic  performance. Do you mean that there is no study IN MALAYSIA ? It is because you have mentioned many similar studies in previous two paragraphs.

2. Please double check whether the root mean square error of approximation (RMSEA) is 0.3. It is because the model is not fit if RMSEA value is greater than 0.08. (From lines 177-178)

3. You state that there is no relationship between environmental factors and academic performance. Even though you give many other studies to support the presence of relationship, you do not give any reasons to explain why there is no relationship found in your present study (from lines 397-419). Please enrich it.   

Reviewer 3 Report

Thank you for resubmission!

Happy with the updated manuscript!

All the best!

Author Response

Thank you